# A Machine Learning Method for Predicting Corrosion Weight Gain of Uranium and Uranium Alloys

**DOI:** 10.3390/ma16020631

**Published:** 2023-01-09

**Authors:** Xiaoyuan Wang, Wanying Zhang, Weidong Zhang, Yibo Ai

**Affiliations:** National Center for Materials and Service Safety, University of Science and Technology Beijing, Beijing 100083, China

**Keywords:** uranium, uranium alloy, corrosion, machine learning, extra trees, feature selection

## Abstract

As an irreplaceable structural and functional material in strategic equipment, uranium and uranium alloys are generally susceptible to corrosion reactions during service, and predicting corrosion behavior has important research significance. There have been substantial studies conducted on metal corrosion research. Accelerated experiments can shorten the test time, but there are still differences in real corrosion processes. Numerical simulation methods can avoid radioactive experiments, but it is difficult to fully simulate a real corrosion environment. The modeling of real corrosion data using machine learning methods allows for effective corrosion prediction. This research used machine learning methods to study the corrosion of uranium and uranium alloys in air and established a corrosion weight gain prediction model. Eleven classic machine learning algorithms for regression were compared and a ten-fold cross validation method was used to choose the highest accuracy algorithm, which was the extra trees algorithm. Feature selection methods, including the extra trees and Pearson correlation analysis methods, were used to select the most important four factors in corrosion weight gain. As a result, the prediction accuracy of the corrosion weight gain prediction model was 96.8%, which could determine a good prediction of corrosion for uranium and uranium alloys.

## 1. Introduction

Uranium and uranium alloys are irreplaceable structural and functional materials in strategic equipment. However, uranium’s unique 5f^3^6d^1^7s^2^ electron arrangement makes it highly chemically reactive and environmentally sensitive, which makes uranium and uranium alloy key structural components highly susceptible to corrosion during long-term service, which, in severe cases, can affect the function of the components, reduce their life, and even cause the failure of the entire device. The accurate and timely assessment of atmospheric corrosion provides important guidance for the material selection and engineering design for corrosion mitigation [1]. Numerous studies on corrosion have been conducted by various researchers. Kelly et al. [2] investigated the relative oxidation and corrosion rates of U–Nb alloys in different temperatures using X-ray photoelectron spectroscopy and neutral mass spectrometry sputtering depth profiling. Zubelewicz et al. [3] presented a constitutive model of a U–6Nb alloy, including the effects of elasticity, crystal reorientation, phase transformations, and plasticity. Wang et al. [4] investigated the oxidative performance of U–2.5Nb alloys at different temperatures in air using the thick weight and weight gain methods. Wang et al. [5] researched the oxidation kinetics of uranium at different times by using a combination of oxygen depletion and reflectance spectroscopy methods. The above studies were instructive for the data collection and study of corrosion mechanisms of uranium and uranium alloys in this paper.

Accelerated corrosion tests have traditionally been conducted to simulate corrosion behavior in a variety of environments, with various electrochemical techniques widely used to predict the corrosion behavior of metals [6]. Li et al. [7] established a prediction model of uranium oxidation and verified it via a 4-year experiment with kinetics parameters obtained through simulated storage and accelerated experiments. Tom et al. [8] presented a description of the implementation of corrosion products into a predictive corrosion model that can be used for the numerical simulation or empirical prediction of uniform corrosion progression. Wang et al. [9] built a model to predict the corrosion behavior of a low-alloy steel in an acidic NaCl solution by using the gray system theory. Accelerated corrosion tests were conducted to calculate the corrosion rate; the feasibility of the model was proved with the simulation results. Although the traditional accelerated simulation test can effectively reduce the test time, there still tend to be some deviations between the simulation test and the actual experimental results. Therefore, the modeling method based on experimental data has also been widely studied by many researchers [10]. Corrosion data are often incomplete, noisy, nonuniform, and bulky (sparse data density). In addition, the service corrosion scene is complex and variable, presenting a highly nonlinear system that is difficult to achieve with traditional statistical methods [6]. Machine learning is a subfield of artificial intelligence (AI), which allows a computer to learn from data to solve a specific task. It includes a flexible method of fitting functions that provides an inexpensive and accurate simulation process compared to traditional computational methods [11,12,13]. Diao et al. [10] collected corrosion data for steels immersed in seawater and built a corrosion rate prediction model of low-alloy steels by using a random forest algorithm. Additionally, they used the gradient boost decision tree (GBDT) algorithm to conduct a feature reduction. However, the GBDT algorithm is sensitive to outliers, and to prevent abnormal data from affecting the feature selection results, extra trees to perform feature selection can be used. Yuan et al. [14] proposed a machine learning model with characteristic parameter embedding to predict and design γ-U alloys in U–Mo–Nb–Ti–Zr systems by using XGBoost regression and genetic algorithm. Using a random forest algorithm, Pei et al. [1] studied the effect of different factors and gas content on atmospheric corrosion. Mythreyi et al. [15] used the extreme gradient boosting algorithm to predict the corrosion performance of the postprocessing and laser-powder-bed-fused (LPBF) Inconel 718. Researchers use a variety of machine learning algorithms when studying corrosion; therefore, in this paper, we first compare 11 classical machine learning regression algorithms in order to select the most applicable one for our data.

As mentioned above, by using machine learning, the data concerning corrosion could be used to achieve more accurate corrosion evaluations. This research applied machine learning methods to the data obtained from previous studies and evaluated the corrosion weight gain of different uranium and uranium alloys in air. Ten-fold cross validation was used to choose the best algorithms, and a combination of extra trees and Pearson correlation coefficient methods was used to perform the feature selection. Finally, a corrosion weight gain prediction model was built.

## 2. Methods

### 2.1. Corrosion Data and Data Preprocessing

The corrosion data we utilized in this research were obtained from previous experimental studies [4,16,17]. We collected 442 rows of laboratory oxidation corrosion data for 8 uranium and uranium alloys in dry air and wet air. The 4 material properties (i.e., type, impurity, phase number, and phase type) of uranium and uranium alloys, 3 typical environmental features (i.e., medium, temperature, and pressure), and corrosion time were also recorded. The corrosion of uranium in air is mainly an oxidation reaction:(1)$$U+\left(\frac{2+x}{2}\right)O_{2}=UO_{2+x}$$
(2)$$U+\left(2+x\right)H_{2}O=UO_{2+x}+\left(2+x\right)H_{2}$$

From the above equation, it can be seen that the overall mass of the sample would increase after being corroded, so the weight gain was used as an output to measure the corrosion process. As shown in Table 1, each corrosion data contained 8 input features and 1 output feature. Furthermore, we employed data preprocessing, removed duplicates and anomalies, and interpolated the data with missing values. We numerated the features whose values were textual, thus, converting categorical features into quantitative variables. For the feature “Type”, we denoted it with “m.n”, where the values of m were 1 and 2 for uranium and uranium alloys, respectively, n for swelling when m = 1, and n for the alloy composition when m = 2. For the feature “Phase_type”, we denoted it with “p.q”, where the values of p are the number of phases and q represents the specific phase. For the feature “Medium”, 1 was for dry air and 2 for humid air. As shown in Figure 1, we plotted the scatter plot of weight gain with respect to time. It can be seen that the oxidation kinetic curves of uranium and the uranium alloys were different due to different factors, such as alloy composition and temperature.

### 2.2. Feature Selection

Feature selection refers to the selection of the most critical features from original features to reduce the dimensionality of a dataset. This can eliminate redundant and less relevant features, mitigating dimensional issues and improving the performance of machine learning models. Feature selection methods are mainly divided into filtered selection, wrapped selection, and embedded selection [18]. In this research, a combination of the extra trees algorithm and Pearson correlation analysis were used to perform the feature selection. The extra trees algorithm is an ensemble learning method in which each decision tree is constructed from the raw training dataset. Each tree randomly selects k features, each feature randomly selects a split node, and then a score for each split node is calculated based on some mathematical metrics (e.g., the Gini index); the node with the highest score is selected as the final split node [19]. This random feature selection makes the randomness of each sub model greater, which suppresses the overfitting of the whole model. In constructing the forest, the normalized total reduction was calculated for each feature using the Gini coefficient, which is the relative importance of that feature. The Pearson correlation method was used to measure the correlation between any two features, and had a value between −1 and 1. A higher Pearson correlation coefficient value indicated a higher correlation between the two variables. Only one of the highest correlation features was selected as an input feature to reduce unnecessary information [20]. Based on the results generated with the two methods mentioned above, several key features could be selected from the original feature set.

### 2.3. Modeling Process

In this research, the processed dataset was divided into a training set and a validation set with various division ratios (i.e., the percentages of the training set were 40%, 50%, 60%, 70%, 80%, and 90%), where the training set was used in the model training phase to estimate the parameters in the model and the validation set was used in the model evaluation phase to verify the predictive accuracy of the model. Then, the predictive performance of 11 classic machine learning algorithms (linear regression, decision tree, extra trees, random forest regression (RFR), kernel ridge regression (KRR), K nearest neighbors (KNN), AdaBoost, gradient boost, bagging, support vector regression (SVR), and light gradient boosting (LGB)) in corrosion weight gain prediction was first compared. After that, several of the relatively well-performing models were then compared after a ten-fold cross-validation [21], and the algorithm with the highest prediction accuracy was selected for subsequent modeling studies. Secondly, feature methods were used to reduce redundant features. Finally, corrosion weight gain prediction models with and without feature selection were established and compared.

The machine learning algorithms used in this research were implemented in Python V3.9 with the Scikit-learn V1.1.2 library. The parameters used to measure the accuracy of the model were as follows:

The coefficient of determination (*R*^2^) is a standard measurement of how well a model fits the data, and it measures the closeness between the observed values and the fitted regression line. The root mean square error (RMSE) is a standard way to quantify the overall error of a regression model, evaluating the deviation between the predicted values and true values. Their specific equations were as follows:(3)$$R^{2}=1-\frac{\sum_{i=1}^{n}{\left(y_{i}-{\hat{y}}_{i}\right)}^{2}}{\sum_{i=1}^{n}{\left({\hat{y}}_{i}-{\bar{y}}_{i}\right)}^{2}+\sum_{i=1}^{n}{\left(y_{i}-{\hat{y}}_{i}\right)}^{2}}$$
(4)$$RMSE=\sqrt{\frac{1}{n}\overset{n}{\underset{i=1}{\sum }}{\left(y_{i}-\hat{y_{i}}\right)}^{2}}$$
where $y_{i}$ denotes the actual value, ${\hat{y}}_{i}$ represents the predicted value, and $\;{\bar{y}}_{i}$ is the mean value of the output.

## 3. Results and Discussion

### 3.1. Comparison of Different Algorithms

The percentages of the training sets to the total datasets were set to 40%, 50%, 60%, 70%, 80%, and 90%, respectively. As shown in Figure 2, the RFR model, extra trees model, gradient boost model, decision tree model, and bagging model had better prediction performance on the validation set than others with different training set division ratios. At the same time, the more data in the training set, the higher the prediction accuracy of most models. In general, the amount of data determines, to some extent, the amount of information it contains. Therefore, in the absence of overfitting, more training data were usually useful for the model to exploit potential relationships that exist in the input features and target attributes. Consequently, a training set ratio of 90% was employed in this research.

In addition, a ten-fold cross-validation was used for the five models mentioned above to reduce the effect of overfitting in the nonlinear regression. As depicted in Figure 3, the extra trees algorithm had the best predictive performance in the ten-fold cross-validation. Thus, the extra trees algorithm was employed in this research. Additionally, for the extra trees algorithm, the maximum RMSE in the ten-fold cross-validation was 1.858, which was more than twice the final average RMSE. Therefore, it can be seen that the ten-fold cross-validation greatly reduced the uncertainty in the selection of the validation set.

### 3.2. Results of Feature Selection

The extra trees method was first used to estimate the correlation between the input features and corrosion weight gain, with the relative importance of each feature shown in Figure 4. To ensure that important features were not lost, we selected features with a relative importance value higher than 0.03 (i.e., the type, impurity, phase_number, and phase_type of material factors and the temperature of environmental factors) as inputs to the model. Additionally, it can be seen that the temperature, type, and impurity were the three most important factors affecting the corrosion of uranium and uranium alloys. Different material types have different corrosion mechanisms, and, in general, alloys have better corrosion resistance than original metals. The corrosion resistance of uranium alloys is closely related to the type of alloying elements, especially to the content of solute elements in uranium alloys. The temperature is also an important factor that affects the corrosion behavior of metals. At low temperatures, the oxidation rate of uranium or uranium alloys is slow, and the initial duration of the reaction is long, while at high temperatures, the oxidation rate is very fast and the initial duration of the reaction is usually short. The number of phases and the type of phase in the microstructure of the uranium alloy are also important factors influencing corrosion performance. The reaction medium and ambient pressure also affect the corrosion behavior, but the data used in this research were all corrosion data in air, so the importance of the characteristics of the medium and pressure was low.

As can be seen in Figure 5, several pairs of input features with strong correlations (i.e., the value of the Pearson correlation coefficient was greater than 0.90) were indicated in red. Among the above selected input features, three pairs of features (i.e., the type and phase_number the type and phase_type, and the phase_number and phase_type) were marked. Simply selecting one feature from each pair of the relevant features mentioned above would give enough information [22]. Typically, features with larger values of relative importance (Figure 3) correlate more with the target attribute (i.e., corrosion weight gain). Therefore, the Type was selected among the three pairs of material features mentioned above. Finally, features, including the Type, Impurity, Temperature, and Time, were selected as the input features of the corrosion weight gain prediction model for uranium and uranium alloys.

### 3.3. Evaluation Results

The performance of the corrosion weight gain prediction models for uranium and uranium alloys with and without feature selection is shown in Figure 6. The green and red lines represent the true and predicted values of the corrosion weight gain for the validation set data, respectively. The *x*-axis represents the samples in the validation set, where each sample contains seven input features and the *y*-axis represents the weight gain, which is the output corresponding to each sample. The values of RMSE and R^2^ of the two models are given in Table 2. Obviously, the R^2^ of both models exceeded 90%, and the RMSE was below 0.7. In addition, by using the out-of-tree and Pearson correlation coefficient methods, R^2^ improved by 0.037% and RMSE decreased by 0.118 compared to the model without the feature selection method. Therefore, the prediction accuracy of the model could be improved by performing feature selection. However, the feature selection also removed the phase organization type, which had an important effect on corrosion behavior, probably because the data in this study were not complex enough and the phase organization type was relatively single. Both models, with and without feature selection, had a satisfactory prediction performance. The good prediction accuracy of the model showed that the corrosion weight gain prediction model should have mastered the influence law of each input feature and the effect of their interaction on the corrosion weight gain. Additionally, the hyperparameters of the model were listed in Table 3.

## 4. Conclusions

In this research, we compared 11 classic machine learning models for the corrosion weight gain prediction of uranium and uranium alloys. The extra trees model, which had the highest prediction accuracy, was selected for predicting corrosion. The model was trained using the following features: type, impurity, phase_number, phase_type, medium, temperature, pressure, and time. Then, a feature selection was performed using the extra trees and Pearson methods, with which redundant features were eliminated. It was found that the prediction accuracy of the model after performing the feature selection was 96.8%, which was a 3% improvement over the previous one, and was able to predict the corrosion data well. Based on the above results, machine learning methods can make sufficient use of corrosion data to determine predictions about corrosion behavior, providing an effective way for performing corrosion research. However, a limitation of this study was that this corrosion weight gain prediction model was only applicable to corrosion data similar to the data in this paper. In our future research, we aim to gather new data that are “unknown” to the model during the training, optimization and validation phases, to create an unbiased evaluation of models, and, if possible, to obtain richer data to further optimize the model for the better prediction of corrosion behavior.

## Figures and Tables

**Figure 1 materials-16-00631-f001:**
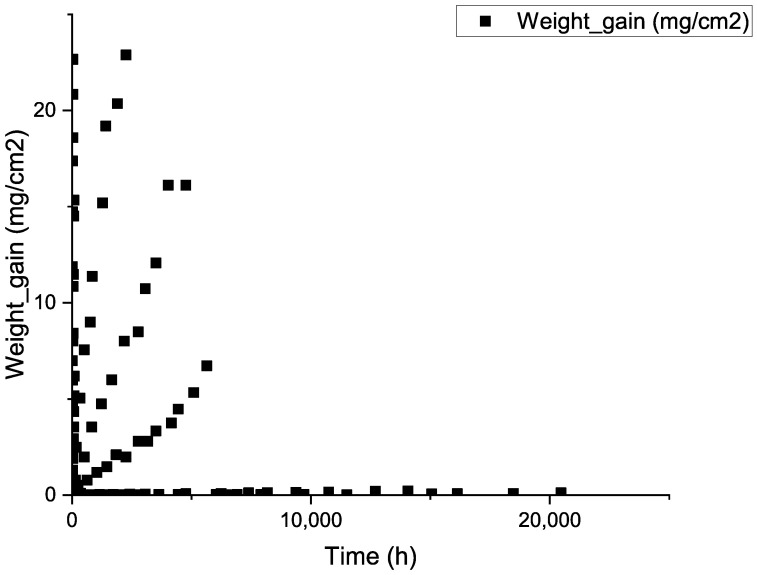
Scatter plot of corrosion weight gain over time for initial data.

**Figure 2 materials-16-00631-f002:**
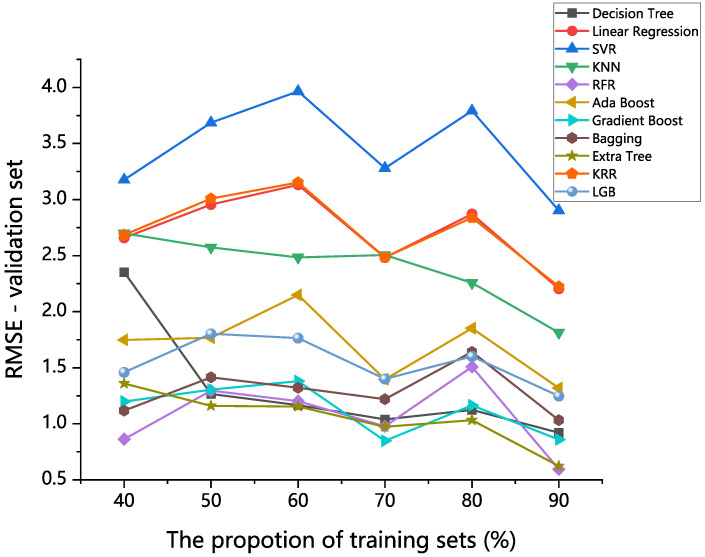
Predictive performance of the decision tree, linear regression, SVR, KNN, KRR, RFR, AdaBoost, gradient boost, bagging, extra trees, and LGB algorithms in different training set division ratios.

**Figure 3 materials-16-00631-f003:**
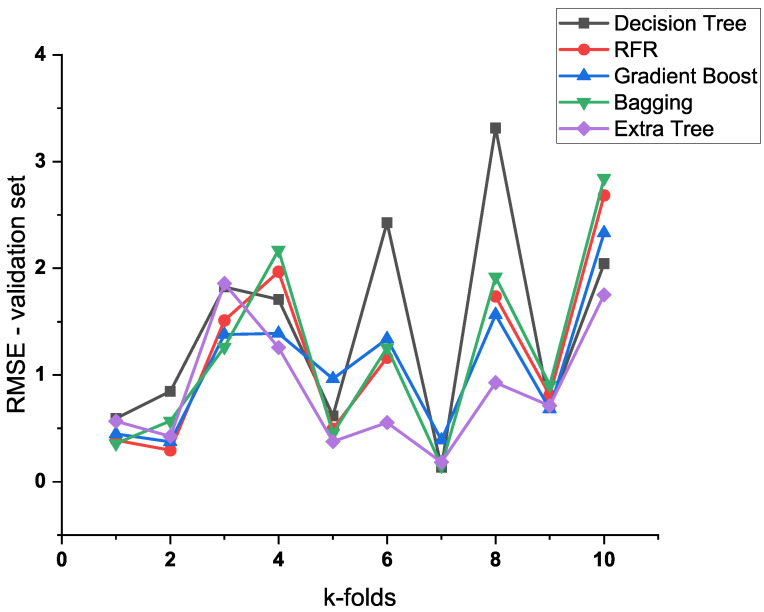
Ten-fold cross-validation comparison of RFR, extra trees, gradient boost, decision tree, and bagging algorithms.

**Figure 4 materials-16-00631-f004:**
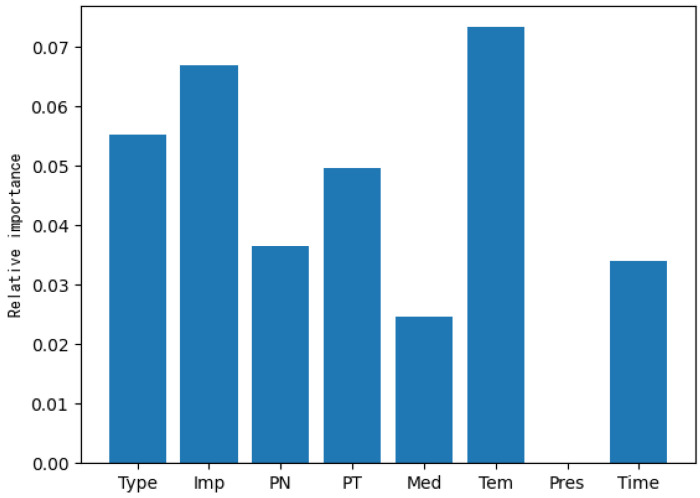
The relative importance of input features with respect to corrosion weight gain evaluated with the extra trees method.

**Figure 5 materials-16-00631-f005:**
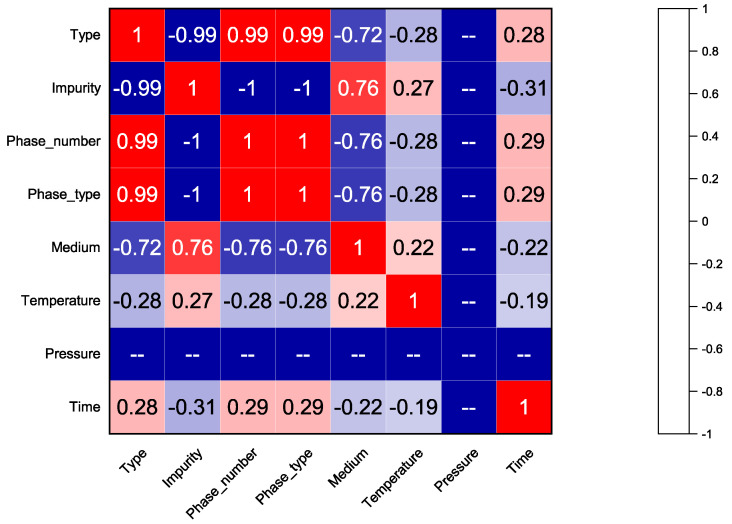
The Pearson correlation map of input features.

**Figure 6 materials-16-00631-f006:**
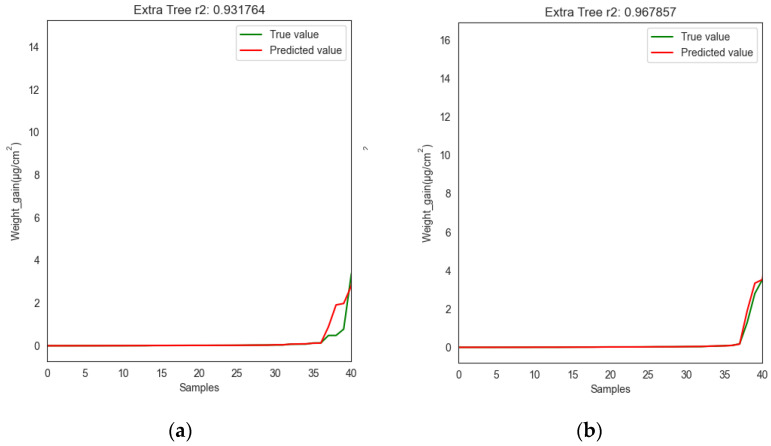
Predictive performance of the models (**a**) with and (**b**) without feature selection.

**Table 1 materials-16-00631-t001:** Description of features in corrosion data.

Features		Unit	Data Range
Material	Type	Data	1–3
	Impurity	%	0.0005–0.002
	Phase_number	Number	1,2
	Phase_type	\	1–3
Environmental	Medium	\	1–2
	Temperature	K	323–573
	Pressure	×10^5^ Pa	1.01325
Reaction time	Time	h	0–200,463
Corrosion weight gain	Weight_gain	mg/cm^2^	0–25

**Table 2 materials-16-00631-t002:** RMSE and R^2^ values for corrosion weight gain prediction models with and without feature selection.

Methods	RMSE	R^2^
Without feature selection	0.634	0.931
With feature selection	0.516	0.968

**Table 3 materials-16-00631-t003:** Hyperparameters of the corrosion weight gain prediction model with feature selection.

Hyperparameter	Value
criterion	“squared_error”
splitter	“random”
max_depth	None
min_samples_split	2
min_samples_leaf	1
min_weight_fraction_leaf	0.0
Max_features	1.0
Max_leaf_nodes	None
min_impurity_decrease	0.0
random_state	None
ccp_alpha	0.0

## Data Availability

The data that supported the findings of this research are in the Appendix A. The code for this work is available on GitHub: “https://github.com/Ca11meMaybe/corrosion-prediction”, accessed on 2 January 2023.

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
