# Peer review of "A Machine Learning Method for Predicting Corrosion Weight Gain of Uranium and Uranium Alloys"

_materials, 2023, doi:10.3390/ma16020631_

Round 1

Reviewer 1 Report

Yibo Ai et al worked on predicting the corrosion rate using machine learning. The work has the potential to be published online, however, there are still some shortcomings that need additional attention for improvement before the work can be formally accepted. Below mentioned are my suggestions for improvements:

1. Abstract seems a bit short, please add more insight and knowledge, and details to the abstract

2. Please also add highlights of the main findings in the abstract

3. The introduction part is also very short and lacks the following information

- Inter-relation with the most recent literature 

- details of the most recent and similar research 

- the novelty of the current work

To me, this work seems more like a scientific report rather than a research work. Most of the technical findings are only described rather than discussed in detail.

Author Response

Response to Reviewer 1 Comments

The authors would like to thank the editor and reviewers for their comments which have helped improve the quality of the manuscript. Responses to the reviewer’s comments are presented on a point-by-point basis below. In the revised manuscript, modified text according to the reviewer’s suggestions has been marked by using “Track Changes” function to enable the reviewer to easily find the revisions.

Point: Yibo Ai et al worked on predicting the corrosion rate using machine learning. The work has the potential to be published online, however, there are still some shortcomings that need additional attention for improvement before the work can be formally accepted. Below mentioned are my suggestions for improvements:

Response: Thank you for your warm comments and professional review on our manuscript. As you are concerned, there are several problems that need to be addressed. According to your nice suggestions, we have made extensive corrections to our previous draft, the detailed corrections are listed below.

Point 1: Abstract seems a bit short, please add more insight and knowledge, and details to the abstract

Response 1: Thank you for your comment. We have added some details to the abstract, please refer to line 10-14:

“There have been substantial studies on metal corrosion research. Accelerated experiments can shorten the test time, but there will still be differences with the real corrosion process. Numerical simulation methods can avoid radioactive experiments but it is difficult to fully simulate the real corrosion environment. Modeling of real corrosion data using machine learning methods allows for effective corrosion prediction.”

Point 2: Please also add highlights of the main findings in the abstract.

Response 2: Thank you for your comment. We have supplemented the abstract:

“As an irreplaceable structural and functional material in strategic equipment, uranium and uranium alloys are generally susceptible to corrosion reactions during service, and prediction of corrosion behavior has important research significance. There have been substantial studies on metal corrosion research. Accelerated experiments can shorten the test time, but there will still be differences with the real corrosion process. Numerical simulation methods can avoid radioactive experiments but it is difficult to fully simulate the real corrosion environment. Modeling of real corrosion data using machine learning methods allows for effective corrosion prediction. This research uses machine learning methods to study the corrosion of uranium and uranium alloys in air and established a corrosion weight gain prediction model. 11 classic machine learning algorithms for regression were compared and ten-folds cross validation method was used to choose the highest accuracy algorithm, which was Extra trees algorithm. Feature selection methods, including Extra Trees and Pearson correlation analysis were conducted to select the most important 4 factors on the corrosion weight gain. As a result, the prediction accuracy of the corrosion weight gain prediction model is 96.8%, which can make a good prediction of corrosion for uranium and uranium alloys.”

Point 3: The introduction part is also very short and lacks the following information

- Inter-relation with the most recent literature 

- details of the most recent and similar research 

- the novelty of the current work

Response 3: Thank you for your comment. We have added corresponding information as your suggestion in the introduction, please refer to line 41-42, 65-68, 71-76 and 80-83:

“The above studies are instructive for data collection and study of corrosion mechanisms of uranium and uranium alloys in this paper.”

“And they use Gradient Boost Decision Tree (GBDT) algorithm to conduct feature reduction. However, GBDT is sensitive to outliers, and to prevent abnormal data from affecting the feature selection results, we use extra trees to perform feature selection.”

“Using a random forest algorithm, Pei et al. [15] studied the effect of different factors and gas content on atmospheric corrosion. Mythreyi et al. [16] used Extreme gradient boosting algorithm to predict the corrosion performance of post-processing and laser-powder-bed-fused (LPBF) Inconel 718. Researchers use a variety of machine learning algorithms when studying corrosion, so in this paper we first compare 11 classical machine learning regression algorithms in order to select the most applicable one for our data.”

“Ten-folds cross validation was used to choose the best algorithms and a combination of Extra Trees and Pearson correlation coefficient methods was used to perform feature se-lection. Finally, a corrosion weight gain prediction model was built.”

Point 4: To me, this work seems more like a scientific report rather than a research work. Most of the technical findings are only described rather than discussed in detail.

Response 4: Thank you for your comment. In accordance with your good instructions, we have revised the whole manuscript carefully and tried to avoid any grammar errors. Modified text has been marked by using “Track Changes” function in the manuscript.

Reviewer 2 Report

1. In the introduction section: Include machine learning-related articles and discuss

2. Research gap should be concise

3. Modelling process: How many numbers of the training set and test sets were chosen for this investigation  

4. What are all the input features?

5. Why uranium is chosen for the corrosion test?

Author Response

Response to Reviewer 2 Comments

The authors would like to thank the editor and reviewers for their comments which have helped improve the quality of the manuscript. Responses to the reviewer’s comments are presented on a point-by-point basis below. In the revised manuscript, modified text according to the reviewer’s suggestions has been marked by using “Track Changes” function to enable the reviewer to easily find the revisions.

Point 1: In the introduction section: Include machine learning-related articles and discuss.

Response 1: Thank you for your comment.

Point 2: Research gap should be concise.

Response 2: Thank you for your comment. We have modified it in the manuscript, please refer to line 65-83.

Point 3: Modelling process: How many numbers of the training set and test sets were chosen for this investigation.

Response 3: Thank you for your comment. 442 rows of data were used in this research and they were divided into a training set and a validation set with various division ratios (i.e., the percentages of the training set were 40%, 50%, 60%, 70%, 80%, 90%, respectively). Consequently, a training set ratio of 90% was chosen for this investigation, which means 398 data for training set and 44 data for validation set.

Point 4: What are all the input features?

Response 4: Thank you for your comment. All the input features are “Type”, “Impurity”, “Phase_number”, “Phase_type”, “Medium”, “Temperature”, “Pressure” and “Time”. After performing feature selection, the input features are “Type”, “Impurity”, “Temperature” and “Time”. Please refer to line 88-91 and 241-243.

Point 5: Why uranium is chosen for the corrosion test?

Response 5: Thank you for your comment. The explanation for why uranium is chosen for the corrosion test is “Uranium and uranium alloys are irreplaceable structural and functional materials in strategic equipment. However, uranium's unique 5f36d17s2 electron arrangement makes it highly chemically reactive and environmentally sensitive, which makes uranium and uranium alloy key structural components highly susceptible to corrosion during long-term service, which in severe cases will affect the function of the components, reduce their life, and even cause the failure of the entire device. Accurate and timely assessment of atmospheric corrosion provides important guidance for material selection and engineering design for corrosion mitigation.” Please refer to line 25-32.

Reviewer 3 Report

The introduction is very well written, makes a comprehensive literature overview and a clear introduction of the problem.

Here are some remarks:

1) Usually, when metals corrode, they tend to lose mass in terms of corrosion products. Hence, one of the most well-known methods to evaluate corrosion is called “Weight loss”. In the case of uranium and its alloys, the authors have used “Weight gain” instead of “Weight loss”. Therefore, the authors should explain why in the case of uranium, it should be used “weight gain” instead of “weight loss”?

2) Page 2, section 2.1: “For the features whose values are textual, we numerate them.” The authors have explained what they have done in words commonly used by materials scientists, and rightly so, but I suggest that they add the terms generally used in machine learning works. Therefore, I would suggest the following sentence: “For the features whose values are textual, we numerate them, thus converting categorical features into quantitative variables.”

3) Page 2, section 2.1: In order to better understand the problem, the authors should include a visual representation of the data. Perhaps, an histogram (or more than one) with the representation of the “Corrosion weight gain” for the different alloys.

4) Page 3, section 2.2: “It can eliminate redundant and irrelevant features, mitigating dimensional disasters and improving the performance of machine learning algorithms.” I suggest that the authors use a softer language, since in statistics we deal with probabilities and not certainties. Therefore, my suggestion would be: “It can eliminate redundant and less relevant features, mitigating dimensional issues and improving the performance of machine learning models”.

5) Page 3, section 2.3: “the processed dataset was divided into a training set and a test set with various division ratios (i.e., the percentages of the training set were 40%, 50%, 60%, 70%, 80%, 90%, respectively), where the training set is used in the model training phase to estimate the parameters in the model and the test set is used in the model evaluation phase to verify the predictive accuracy of the model” In this early phase, what the authors call a test set, is still a validation set, albeit with different percentages of the whole dataset. A test set is a final holdout dataset (or new data) containing data never “seen” by the model during the training and validation phases, which can be used to test the best and final model. Therefore, in the sentence above, instead of “test set”, it should be a “validation set”.

6) Page 4, Figure 1: Instead of “RMSE of test set”, it should be “RMSE - Validation set”

7) Page 4, Figure 1: The definition of the Figure should be improved. If the graphic was created with origin, the authors might test changing the way they do the copy-paste of the figure.

8) Page 4, section 3.1: The paragraph should be divided into two, the first paragraph refers to Figure 1, and the second would refer to Figure 2. The second would start at: “In addition, ten-fold cross-validation was used for the five models mentioned above to reduce the effect of overfitting in nonlinear regression. As depicted in Fig. 2, the Extra Tree algorithm had the best predictive performance in ten-fold cross-validation. Thus, the Extra Tree algorithm was employed in this research.”

9) Page 5, Figure 2: “Validation times” Do the authors mean the number of k-folds? If so, the authors can write “k-folds” or “nº of k-folds”.

10) Page 7, section 3.3: “For datasets with different input features, both models have better prediction performance.” I failed to understand what the authors mean. Can the authors explain this sentence? Or, do the authors simply mean that: “Both models, with and without feature selection, have a satisfactory prediction performance”.

11) Page 7, section 3.3: “The model without feature selection can be used in the case of rich data features, while the model with feature selection is suitable for the case of less rich data features (e.g., data with only alloy types but no phase organization types)” According to these results the model with feature selection should always be preferred. If more differentiated data is obtained in the future, the authors should test again this premise. Therefore, this comment should be removed, or better explained.

12) Page 7, Figure 5: What do the authors mean by samples? This should also be explained in the article.

13) Page 7, Table 2: The uncertainties from cross-validation should be reported. It could be twice (approximate t-critical value) the standard deviation of the mean for the results of each of the 10 folds, for R2 and RMSE.

14) Page 8, Data Availability Statement: “The data that support the findings of this research are available from the corresponding author, Y.B.A. and W.D.Z. upon reasonable request.” This might be allowed by the journal, but I strongly suggest that the authors make the data openly available, perhaps through Mendeley Data. The work is in a very niche area of corrosion science and it does not involve commercial interests. Therefore, by making the data openly available, it will incentivize other authors to do research this field, which could potentially lead to more citations, making the authors’ work even more well-known, thus creating even more benefits for them long-term.

15) A machine learning work can only be truly evaluated by examining the code, therefore, the code should be made openly available online, through github, for example.

16) The hyperparameters of the best performing method (Extra Trees) should be specified.

17) The authors didn’t include a final test (after choosing the best method according to 10-fold cross-validation) with an holdout sample. Therefore, they should gather new data, “unknown” to the model during the train, optimization and validation phases, in order to make an unbiased evaluation of the model. If this is not possible or it is too cumbersome for the present work, the authors should clearly state this in the article (perhaps, in the conclusions), and discuss the next steps they are planning to undertake in order to check the developed machine learning methodology?

Overall, the article seems consistent, important to the literature and well executed. However, the previous remarks should be addressed before the article is able to be considered for publication.

Author Response

Response to Reviewer 3 Comments

The authors would like to thank the editor and reviewers for their comments which have helped improve the quality of the manuscript. Responses to the reviewer’s comments are presented on a point-by-point basis below. In the revised manuscript, modified text according to the reviewer’s suggestions has been marked by using “Track Changes” function to enable the reviewer to easily find the revisions.

Point: The introduction is very well written, makes a comprehensive literature overview and a clear introduction of the problem.

Here are some remarks:

Response: Thank you for your warm comments and careful review on our manuscript. We have read the comments carefully and revise our manuscript. In accordance with your good instructions, we have revised the whole manuscript carefully and tried to avoid any grammar errors. We have modified the manuscript accordingly, and detailed revisions are listed below point by point:

Point 1: Usually, when metals corrode, they tend to lose mass in terms of corrosion products. Hence, one of the most well-known methods to evaluate corrosion is called “Weight loss”. In the case of uranium and its alloys, the authors have used “Weight gain” instead of “Weight loss”. Therefore, the authors should explain why in the case of uranium, it should be used “weight gain” instead of “weight loss”?

Response 1: Thank you for your comment. An explanation has been added to the manuscript to explain why using “weight gain” for uranium and its alloys, please refer to line 91-96:

” The corrosion of uranium in air is mainly an oxidation reaction:

(1)

(2)

From the above equation, it can be seen that the overall mass of the sample will increase after being corroded, so the weight gain is used as an output to measure the corrosion process.”

Point 2: Page 2, section 2.1: “For the features whose values are textual, we numerate them.” The authors have explained what they have done in words commonly used by materials scientists, and rightly so, but I suggest that they add the terms generally used in machine learning works. Therefore, I would suggest the following sentence: “For the features whose values are textual, we numerate them, thus converting categorical features into quantitative variables.”

Response 2: Thank you for your comment. We have revised the sentence as your suggestion.

Point 3: Page 2, section 2.1: In order to better understand the problem, the authors should include a visual representation of the data. Perhaps, an histogram (or more than one) with the representation of the “Corrosion weight gain” for the different alloys.

Response 3: Thank you for your comment. We have added a Scatter plot of corrosion weight gain over time for initial data and corresponding explanation in our manuscript, please refer to line 105-111:

“As shown in Fig. 1, we plotted the scatter plot of weight gain with respect to time. It can be seen that the oxidation kinetic curves of uranium and uranium alloys are different due to different factors such as alloy composition and temperature.

Figure 1. Scatter plot of corrosion weight gain over time for initial data.”

Point 4: Page 3, section 2.2: “It can eliminate redundant and irrelevant features, mitigating dimensional disasters and improving the performance of machine learning algorithms.” I suggest that the authors use a softer language, since in statistics we deal with probabilities and not certainties. Therefore, my suggestion would be: “It can eliminate redundant and less relevant features, mitigating dimensional issues and improving the performance of machine learning models”.

Response 4: Thank you for your comment. We have modified it as your suggestion.

Point 5: Page 3, section 2.3: “the processed dataset was divided into a training set and a test set with various division ratios (i.e., the percentages of the training set were 40%, 50%, 60%, 70%, 80%, 90%, respectively), where the training set is used in the model training phase to estimate the parameters in the model and the test set is used in the model evaluation phase to verify the predictive accuracy of the model” In this early phase, what the authors call a test set, is still a validation set, albeit with different percentages of the whole dataset. A test set is a final holdout dataset (or new data) containing data never “seen” by the model during the training and validation phases, which can be used to test the best and final model. Therefore, in the sentence above, instead of “test set”, it should be a “validation set”.

Response 5: Thank you for your comment. “test set” has been revised as “validation set” throughout the manuscript.

Point 6: Page 4, Figure 1: Instead of “RMSE of test set”, it should be “RMSE - Validation set”.

Response 6: Thank you for your comment. We have modified it.

Point 7: Page 4, Figure 1: The definition of the Figure should be improved. If the graphic was created with origin, the authors might test changing the way they do the copy-paste of the figure.

Response 7: Thank you for your comment. Figure 1, Figure 2 and Figure were created with origin. At first we had a problem with the origin software, so the figure was a screenshot, now we have redownloaded origin and imported the figures.

Point 8: Page 4, section 3.1: The paragraph should be divided into two, the first paragraph refers to Figure 1, and the second would refer to Figure 2. The second would start at: “In addition, ten-fold cross-validation was used for the five models mentioned above to reduce the effect of overfitting in nonlinear regression. As depicted in Fig. 2, the Extra Tree algorithm had the best predictive performance in ten-fold cross-validation. Thus, the Extra Tree algorithm was employed in this research.”

Response 8: Thank you for your comment. We have modified it according to your advice.

Point 9: Page 5, Figure 2: “Validation times” Do the authors mean the number of k-folds? If so, the authors can write “k-folds” or “nº of k-folds”.

Response 9: Thank you for your comment. It acctually mean the number of k-folds, we have revised it as “k-folds”.

Point 10: Page 7, section 3.3: “For datasets with different input features, both models have better prediction performance.” I failed to understand what the authors mean. Can the authors explain this sentence? Or, do the authors simply mean that: “Both models, with and without feature selection, have a satisfactory prediction performance”.

Response 10: Thank you for your comment. It simply mean that: “Both models, with and without feature selection, have a satisfactory prediction performance”. We have revised this sentence in our manuscript.

Point 11: Page 7, section 3.3: “The model without feature selection can be used in the case of rich data features, while the model with feature selection is suitable for the case of less rich data features (e.g., data with only alloy types but no phase organization types)” According to these results the model with feature selection should always be preferred. If more differentiated data is obtained in the future, the authors should test again this premise. Therefore, this comment should be removed, or better explained.

Response 11: Thank you for your comment. This sentence have benn removed.

Point 12: Page 7, Figure 5: What do the authors mean by samples? This should also be explained in the article.

Response 12: Thank you for your comment. An explanation has been added to the manuscript, please refer to line 254-256:

“The x-axis represents the samples in the validation set, each sample contains 7 input features, and the y-axis represents the weight gain, which is the output corresponding to each sample.”

Point 13: Page 7, Table 2: The uncertainties from cross-validation should be reported. It could be twice (approximate t-critical value) the standard deviation of the mean for the results of each of the 10 folds, for R2 and RMSE.

Response 13: Thank you for your comment. We have added the report to uncertaninties from cross-validation in our manuscript, please refer to line 187-190:

“And for the Extra trees algorithm, the maximum RMSE in the ten-fold cross-validation is 1.858, which is more than twice the final average RMSE. Therefore, it can be seen that the ten-fold cross-validation greatly reduces the uncertainty in the selection of the validation set.”

Point 14: Page 8, Data Availability Statement: “The data that support the findings of this research are available from the corresponding author, Y.B.A. and W.D.Z. upon reasonable request.” This might be allowed by the journal, but I strongly suggest that the authors make the data openly available, perhaps through Mendeley Data. The work is in a very niche area of corrosion science and it does not involve commercial interests. Therefore, by making the data openly available, it will incentivize other authors to do research this field, which could potentially lead to more citations, making the authors’ work even more well-known, thus creating even more benefits for them long-term.

Response 14: Thank you for your comment. We have upload the data through Mendeley Data, but it is in moderation. After being accept, it can be seen online. And we have submitted the data to the supplementary material.

Point 15: A machine learning work can only be truly evaluated by examining the code, therefore, the code should be made openly available online, through github, for example.

Response 15: Thank you for your comment. The code has been available on github and we have revised the Data Availability Statement:

Point 16: The hyperparameters of the best performing method (Extra Trees) should be specified.

Response 16: Thank you for your comment. The hyperparameters of the best performing method have been listed in our manuscript, please refer to line 267-268 and 274-288:

“And the hyperparameters of the model were listed in table 3.

Table 3. Hyperparameters of the corrosion weight gain prediction model with feature selection.

Hyperparameter

Value

criterion

“squared_error”

splitter

“random”

max_depth

None

min_samples_split

2

min_samples_leaf

1

min_weight_fraction_leaf

0.0

Max_features

1.0

Max_leaf_nodes

None

min_impurity_decrease

0.0

random_state

None

ccp_alpha

0.0

Point 17: The authors didn’t include a final test (after choosing the best method according to 10-fold cross-validation) with an holdout sample. Therefore, they should gather new data, “unknown” to the model during the train, optimization and validation phases, in order to make an unbiased evaluation of the model. If this is not possible or it is too cumbersome for the present work, the authors should clearly state this in the article (perhaps, in the conclusions), and discuss the next steps they are planning to undertake in order to check the developed machine learning methodology?

Response 17: Thank you for your comment. It is too cumbersome for the present work to get new data so we added a disscussion in the conclusions, plesease refer to  line 299-304:

“However, a limitation of this study is that this corrosion weight gain prediction model is only applicable to corrosion data similar to the data in this paper. In our future research, we aim to gather new data that “unknown” to the model during the train, optimization and validation phases, to make an unbiased evaluation of the model, and if possible, to obtain richer data to further optimize the model for better prediction of corrosion behavior.”

Round 2

Reviewer 1 Report

The manuscript is in good shape and can be published online.

Reviewer 3 Report

The authors have a made a good job answering all the questions raised in this reviewer. Therefore, taking into account the quality of the work and the manuscript, together with its impact in the corrosion science community and contribution to solve a societal problem, I recommend the article to be accepted in the present form.

Congratulations!

PS: Can the authors please confirm if the Github was made public in the definitions? Thank you.